# A Systematic Literature Review on the Application of Automation in Logistics

Bárbara Ferreira [1] and João Reis [2,*]

1   Higher Institute of Management and Administration of Santarém (ISLA), 2000-241 Santarém, Portugal
2   Industrial Engineering and Management, Faculty of Engineering, Lusófona University and RCM2+, 1749-024 Lisbon, Portugal
*   Correspondence: joao.reis@ulusofona.pt

**Abstract:** *Background*: in recent years, automation has emerged as a hot topic, showcasing its capacity to perform tasks independently, without constant supervision. While automation has witnessed substantial growth in various sectors like engineering and medicine, the logistics industry has yet to witness an equivalent surge in research and implementation. Therefore, it becomes imperative to explore the application of automation in logistics. *Methods*: this article aims to provide a systematic analysis of the scientific literature concerning artificial intelligence (AI) and automation in logistics, laying the groundwork for robust and relevant advancements in the field. *Results:* the foundation of automation lies in cutting-edge technologies such as AI, machine learning, and deep learning, enabling self-problem resolution and autonomous task execution, reducing the reliance on human labor. Consequently, the implementation of smart logistics through automation has the potential to enhance competitiveness and minimize the margin of error. The impact of AI and robot-driven logistics on automation in logistics is profound. Through collaborative efforts in human–robot integration (HRI), there emerges an opportunity to develop social service robots that coexist harmoniously with humans. This integration can lead to a revolutionary transformation in logistics operations. By exploring the scientific literature on AI and automation in logistics, this article seeks to unravel critical insights into the practical application of automation, thus bridging the existing research gap in the logistics industry. *Conclusions*: the findings underscore the impact of artificial intelligence and robot-driven logistics on improving operational efficiency, reducing errors, and enhancing competitiveness. The research also provided valuable insights into the applications of various automation techniques, including machine learning and deep learning, in the logistics domain. Hence, the study's insights can guide practitioners and decision makers in implementing effective automation strategies, thereby improving overall performance and adaptability in the dynamic logistics landscape. Understanding these foundations can pave the way for a future where automation and human expertise work hand in hand to drive logistics toward unparalleled efficiency and success.

**Keywords:** logistic; automation; artificial intelligence; machine learning; deep learning; systematic literature review

## 1. Introduction

Intelligent systems have significantly transformed the logistics landscape, primarily through the adoption of automation, which has proven effective in mitigating failure and accidents [1]. Artificial intelligence (AI) has emerged as a reliable and cost-effective solution, empowering businesses to address uncertainty and efficiently manage complex algorithms [2]. While automation has been extensively studied in computer science [3], engineering [4], mathematics [5], and medicine [6], logistics in scientific research still falls short when compared with these domains. To justify our previous argument, we conducted a search on the Scopus database during the final quarter of 2023. Employing the search term "artificial intelligence" in the title, abstract, and keywords, we observed

substantial representation in various academic disciplines. Specifically, the field of computer sciences demonstrated the highest prevalence at 37%, followed by engineering at 16.4%, mathematics at 14%, and medicine at 5.4%. When restricting the analysis to the most recent five-year period, a slight shift becomes apparent. During this timeframe, the representation of "artificial intelligence" in the field of computer science amounts to 28.3%, followed by engineering at 16.2%, medicine at 8.9%, and mathematics at 8.7%. In contrast, the occurrence of "artificial intelligence" within disciplines such as materials science and management was comparatively minimal, accounting for 2.1% and 1.8%, respectively. In the last 5 years, materials science has presented 3% and management has been around 2%. Furthermore, we expanded our investigation by incorporating dual search terms with no time restriction. The combination of "artificial intelligence" and "engineering" yielded 45,050 manuscripts, while "artificial intelligence" and "computer sciences" resulted in 38,234 manuscripts. Similarly, the search pairing "artificial intelligence" with "medicine" generated 13,394 manuscripts, whereas the intersection of "artificial intelligence" and "logistics" produced a relatively modest 7,333 manuscripts. Some notable studies, such as Woschank et al. [7] research on AI in smart logistics and Iyer et al. [1] work on intelligent transportation, have shed light on this issue. "Smart logistics" encompasses the application of advanced technologies [8], data analysis [9], and automation [10] in the logistics domain, aiming to optimize supply chain processes [11]. This integration involves interconnected systems and devices enabling real-time monitoring, analysis, and decision making in logistics operations [12] By utilizing automation and digital technologies, smart logistics strives to enhance supply chain performance, visibility, and responsiveness, resulting in improved productivity, cost efficiency, and heightened customer satisfaction [13]. Automation facilitates this transformation by enabling intelligent decision-making algorithms and machine learning models that optimize routing, scheduling, and final delivery, thereby ensuring efficient resource allocation, cost reduction, and timely delivery of goods [13]. Overall, the incorporation of automation in logistics plays a pivotal role in the evolution of conventional supply chains into intelligent, interconnected, data-driven systems [14] that underpin the concept of smart logistics. In this context, several articles delve into the potential applications of cutting-edge technologies, such as the AR/VR (augmented reality/virtual reality) technologies within Industry 4.0. In that regard, articles such Machała et al. [15] examine the integration of smart glasses and mobile devices to facilitate augmented reality experiences, thereby expediting work processes and data transfers in developed nations' manufacturing, warehousing, and transportation sectors. It is important to note that this is a single instance illustrating the broader scope of possibilities within this domain.

However, a comprehensive understanding of the impact of automation on logistics and its development still requires further investigation. To bridge this research gap, we conducted a systematic literature review (SLR) to identify research deficiencies and assess the current state of knowledge regarding automation in logistics from a holistic perspective. This review aims to comprehend how automation is currently enhancing the logistics industry and the ongoing developments in this field. Specifically, we address the research question: how does automation contribute to the improvement of the logistics industry? In this context, AI plays a pivotal role by providing systems with the critical capability of cognition, encompassing modeling, representation, and learning of complex behaviors and interactions within a system's components and data. This ability enables autonomous decision making by robots, akin to human-like intelligence. By leveraging machine learning, operational efficiency is achieved, costs are reduced, and the work environment is enhanced, empowering the workforce to create and implement process innovations. Additionally, we discovered that a collaborative human–robot integration (HRI) offers the potential to create social service robots that seamlessly coexist with humans in social settings, catering to the genuine demands and expectations of lay experts.

Accordingly, the purview of this article predominantly centers on examining the utilization of automation, specifically AI, machine learning, and deep learning, within the sphere of the logistics domain. This article scrutinizes the transformative influence of these

technologies on several dimensions of logistics operations, encompassing but not limited to inventory management, supply chain optimization, transportation, and warehouse automation. The primary objective of the article is to furnish a holistic comprehension of the contemporary landscape, challenges, and prospects inherent in the assimilation of automation into the complex web of logistics processes. Particularly, the article deliberately abstains from studying broader subjects detached from the ambit of automation in logistics, such as generic transportation analyses or strategies for logistics management that do not involve automated frameworks.

The following sections of this article are organized as follows: Section 2 outlines the methods employed in this systematic review, including the PRISMA statement (Preferred Reporting Items for Systematic Review and Meta-Analysis Protocol), the SLR process, and content analysis. Section 3 presents the findings from the analyses conducted. Finally, Section 4 discusses the theoretical and managerial contributions derived from this research, illuminating the way forward for automation in the logistics industry.

## 2. Materials and Methods

### 2.1. Search Strategy

We opted to use Elsevier Scopus as the primary database for our keyword search due to its well-established reputation for systematically mapping and reviewing the literature [16,17]. Notably, Scopus boasts extensive coverage of articles across various scientific domains, including business and management, engineering, and computer science [18]. Moreover, many researchers in the fields of automation, AI, and logistics publish their work in this database [7,19]. Our search, conducted on 7 February 2023, involved screening the article title, abstract, and keywords using the following search terms: "Logistic*" and "Automation" and "Artificial Intelligence". This led to the identification of a total of 301 manuscripts. To ensure the inclusion of only high-quality studies, we initially considered all types of studies but limited our selection to journal articles published in the English language. Additionally, we set a timeframe for the research studies from 2019 to 2023, as a growing trend of publications in this field was observed during these years [19]. This step was vital in capturing the most recent and relevant contributions to the domain of automation in logistics.

### 2.2. PRISMA Protocol

The primary objective of this review is to conduct a thorough and comprehensive search of published manuscripts, making an SLR the appropriate approach. One of the primary benefits of SLRs, as outlined in the literature, is their focused and transparent approach [20]. In adherence to this principle, we adopted the PRISMA statement, enabling us to systematically analyze the research methodology step by step. During our investigation of the Web of Science (WoS) database, we observed that Scopus contained a significant number of highly relevant articles within the specific research areas of interest [19]. This fact led us to consider Scopus as our initial choice for selecting a single database. Furthermore, Scopus enjoys a widely recognized reputation as one of the most comprehensive repositories of abstracts and citation research literature globally. Scholars have also affirmed that Scopus stands as one of the largest and most pertinent databases, providing a wealth of high-quality documents pertaining to automation and logistics. While academic search engines like Google Scholar may yield a more extensive collection of documents, our primary goal was to focus on studies with blind peer review. In this regard, Scopus proved to be the optimal choice, aligning with our criteria for obtaining rigorously evaluated and reliable research. In summary, based on its comprehensiveness, relevance, and popularity among scholars, Scopus was deemed the most suitable database for our study, ensuring a robust and thorough examination of the automation and logistics literature. Thus, Scopus was employed as the primary source, facilitating the collection of information pivotal to the construction of this article's conceptual framework. However, this did not preclude the

utilization of supplementary supporting manuscripts from Web of Science (WoS), Google Scholar, and various internet-based platforms as will be seen later by readers.

Figure 1 presents the initial pool of 301 manuscripts identified during the identification phase. To ensure the inclusion of the highest quality articles, we applied a series of filters in the review process. Initially, we excluded non-English manuscripts, as English is widely recognized as the universal language of scientific research, resulting in 291 remaining manuscripts. Next, we applied the "Journal" filter, narrowing the results down to 123 manuscripts. Further refining the selection, we applied the "articles" filter, which left us with 108 articles that met our criteria for high-quality studies. To capture the most recent and relevant contributions to the field of logistics and automation, we applied a time filter, considering only articles published between 2019 and 2023. This five-year time-frame was chosen because it coincided with a notable upsurge in published articles in this area. Throughout the eligibility phase, we examined all chosen manuscripts, subsequently excluding 52 articles inaccessible to us, resulting in a final selection of 17 articles. Although the initial search was conducted solely through Scopus, we recognized the importance of including other articles that offered relevant contributions to our study. Consequently, the search was expanded to 12 additional articles, resulting in a final set of 29 reviews that met our stringent criteria for inclusion. The application of these filters and the inclusion of relevant articles allowed us to focus on a select group of high-quality studies, providing a comprehensive foundation for our review on the topic of logistics and automation.

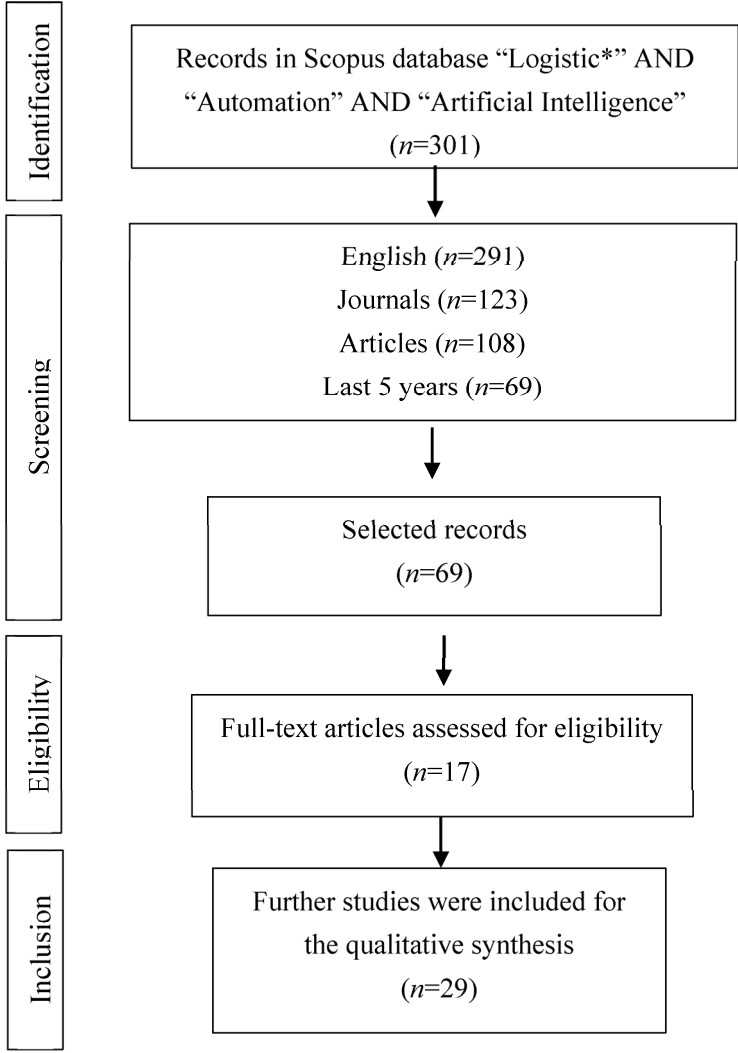

**Figure 1.** PRISMA Protocol.

Hence, in our empirical investigation, the primary selection of manuscripts for analysis in the eligibility phase comprised 17 articles. This core set was further augmented by an additional 12 principal articles, expanding the total set to 29 articles during the inclusion phase (n = 29). The rationale underlying the incorporation of the supplementary 12 articles (from WoS) was to both validate and introduce diverse perspectives, thereby enhancing the comprehensiveness of our discourse and preempting an overly restrictive view with limited deliberation. Illustratively, within the last paragraph of Section 3, the discourse encapsulates the viewpoints of Heinrich et al., representing a subset of the 17 initial authors. Building upon the insights of Heinrich et al., we incorporated the contributions of Knight, which transcended the confines of the former's scope. This approach signifies a comprehensive analysis rooted in the Scopus article corpus, reinforced by the inclusion of authors during the inclusion phase, aiming to bolster or challenge the content derived from the initial selection phase. It is essential to emphasize that the other additional manuscripts (from Google Scholar and various internet-based platforms, e.g., private companies such as DHL) presented in the results maintain a purely supplementary role and do not form part of the core conceptual framework. Their function primarily revolves around providing contextual framing and, in certain instances, presenting illustrative business cases, typified by the inclusion of the DHL case, thus fortifying supplementary practical arguments. This approach serves to enhance the visibility of the presented arguments and establish a trajectory for the future empirical validation of the conceptual framework. Readers of this article seeking comprehensive access to the complete list of articles and their specific role in the analysis process are encouraged to reach out to the corresponding author, who will address any inquiries regarding the article's selection context.

### 2.3. Content Analysis Technique

The data under discussion in this article were synthesized employing the well-established content analysis technique. This specific approach was selected for its capacity to offer a comprehensive view of contemporary phenomena, particularly within qualitative research methodologies [21]. This technique is recognized for its efficacy in scrutinizing non-statistical data [22] and exploring the underlying rationales and methodologies elucidating the "how" of the research question [23]. The application of this technique encompassed a thorough examination of the textual corpus identified earlier, ensuring an examination of each individual article. Throughout this process, ideas were discerned, and keywords and phrases were systematically coded. Subsequently, this initial coding phase facilitated the identification of overarching categories and their respective subcategories. This procedure enabled us to discern patterns within the codes, ultimately unveiling pertinent themes and facilitating the creation of a comprehensive map that provided an overview of the data, thereby revealing emergent patterns and ideas. Following the completion of the mentioned process, we revisited our methodology, cross-examining the identified keywords in the articles to detect any disparities between the results of the content analysis and the most salient terms attributed by the authors. Once similarities were ascertained, we progressed to discuss the outcomes in the subsequent section, with a high degree of confidence that the topics and sub-topics highlighted within the conceptual framework are indeed robust and reliable.

## 3. Results

In this section, we will conduct an in-depth analysis of the descriptive findings related to automation in logistics. Furthermore, we will perform a content analysis of the identified papers to gain a comprehensive understanding of their key insights and contributions.

### 3.1. Artificial Intelligence

AI is a rapidly advancing field in science and engineering, focused on creating intelligent machines and computer programs [24]. Moreover, AI possesses the remarkable ability to automate learning and discovery processes by processing information, allowing

it to handle repetitive, high-volume tasks tirelessly [25]. Through successive learning algorithms, AI is gaining knowledge about patterns and structures in its environment, enabling it to adapt and learn from new experiences. However, there are concerns about potential cyberattacks and biased decisions made by AI systems. To address these risks, the European Union (EU) is encouraging companies to adopt regulatory frameworks for innovation, while safeguarding fundamental values and rights [26]. The transportation and logistics sector, responsible for handling valuable goods worldwide, faces challenges that AI can help overcome. Companies like AI EdgeLabs emphasize the importance of AI in addressing these challenges and improving logistics security [27]. Despite the benefits, AI can also make biased decisions based on historical data. For instance, the male-dominated logistics industry might lead to biased task assignments due to a lack of data related to female work and capacities [28,29]. The potential of AI to enhance the customer journey is significant, aiding product search, lead generation, and customer retention [30,31]. In logistics, AI finds applications in inventory control, stock taking, tracking, forecasting, and back orders, as well as internal logistics and transport processes. Companies like DHL (Dalsey, Hillblom, and Lynn) and Sky Planner leverage AI to improve task completion time predictions and anticipate future behavior [32]. Among various AI technologies, artificial neural networks (ANN) are extensively used. ANN can significantly reduce inventory levels while maintaining customer demand satisfaction probability [33,34]. Additionally, the precision calculations of ANN help anticipate ship-related incidents [35]. The role of AI in analyzing big data from cyber-physical systems (CPS) is also crucial for automating processes and gaining better visibility into logistics systems [7]. CPS is known to be an embedded system that bridges the physical and digital worlds, offering real-time data access and processing services [36]. The cognitive abilities of AI enable it to model and learn complex patterns, making it instrumental in facilitating large-scale CPS and empowering autonomous decision making [37]. Moreover, the innovative cyber-physical systems developed by institutions like RMIT University aim to enhance human–machine collaboration and build trustworthy autonomy by leveraging cutting-edge cognitive processing and machine learning techniques [38]. These advancements have the potential to revolutionize logistics and various other industries, enabling them to tackle complex challenges and optimize operations. By embracing AI, while being mindful of potential biases, the logistics sector can unlock substantial benefits in efficiency, accuracy, and decision making, paving the way for a more robust and productive future. In the next subsection, we will explore topics such as machine learning and deep learning, followed by a comprehensive analysis of robot-driven logistics.

### 3.1.1. Machine Learning

To enhance efficiency in logistics, utilizing machine learning is a suitable and powerful approach for analyzing various prediction and decision problems [39]. Machine learning excels in solving classes of predictions and decisions where traditional analytical or programmatic methods may not efficiently handle domains with data-driven experiences, thus automatically improving the overall efficiency of solutions [40]. Moreover, machine learning can be employed to extract valuable insights [41], such as identifying rare but significant customer needs [42].

Data-driven AI combines machine learning techniques like supervised, unsupervised, and reinforcement learning with large-scale data analysis to identify risks and market trends and ease traffic congestion [1]. In that regard, supervised learning involves learning from labeled input–output data and has been extensively used in logistics, utilizing methods like multilayer perceptron artificial neural networks, gradient boosting trees, random forest, naive Bayes, and logistic regression [43,44]. On the other hand, unsupervised learning uses non-labeled data for training, employing dimensionality reduction and clustering to organize datasets and detect structure in the data [43,44]. For example, in logistics, unsupervised learning (clusters) can be used to address challenges in intermodal transportation, where no straightforward routes to the destination exist. Lastly, reinforcement learning

is another valuable approach, allowing an algorithm to learn how to act in an unfamiliar environment to maximize rewards. In logistics, reinforcement learning can be utilized to address issues like demand uncertainty, where fluctuations caused by external factors can lead to unexpected increases or decreases in demand [43,45].

Machine learning (ML) is providing relevant contributions in known domains, such as the medical field. In this case, ML using recurrent neural networks (RNN) can be applied for stroke diagnosis and tracking, leveraging their ability to recognize patterns and sequences to predict likely scenarios [46]. In logistics, machine learning is similarly providing relevant contributions by testing for anomaly detection, successfully producing accurate discriminative models integrated into smart logistics management systems with high accuracy levels [47]. For instance, predictive maintenance software utilizing machine learning and deep learning algorithms, such as Presenso, are significantly benefiting logistics companies by optimizing maintenance processes and operational efficiencies [7]. Under-sampling machine learning models can amplify the anticipated profit of back-order decisions [48]. Therefore, machine learning enables predicting future events, like traffic, leading to increased operational efficiency, cost reduction, and fostering a work environment that encourages innovation and process improvements [49,50]. It also paves the way for autonomous acting, decision-making processes, and the achievement of autonomous configuration [51].

Overall, embracing machine learning in logistics is presenting a transformative opportunity, empowering businesses to make more informed decisions, improve overall performance, and adapt swiftly to evolving challenges and opportunities in the dynamic logistics landscape.

### 3.1.2. Deep Learning

Deep learning is a specialized subset of AI techniques and falls under the umbrella of machine learning. Its primary goal is to explore the intricate layers of non-linear information processing, enabling the analysis and classification of supervised and unsupervised features and patterns. Both machine learning and deep learning algorithms heavily rely on the quantity of data used for training. As machine learning algorithms approach consistency with an adequate amount of training data, deep learning algorithms exhibit enhanced performance as the volume of data increases. In the present scenario, deep learning finds widespread applications, exemplified by Google's voice and image recognition capabilities, HBO and Amazon's recommendation engines, Apple's Siri, automated email, and text reply processing, as well as chatbots [52]. Moreover, deep learning facilitates image detection, empowering machines to identify various objects and transmit their findings to a cloud-based server [53]. Notably, deep learning also plays a significant role in predictive maintenance, as highlighted by Wang and Wang [54]. They discuss an array of cutting-edge algorithms in this field, such as convolutional neural networks, deep belief networks, and long short-term memory models. These sophisticated algorithms are considered pivotal to the future success of companies [54].

Considering the above, the literature also discusses convolutional neural networks (CNN), an effective classification algorithm that assigns weights and biases to distinct objects in an image, thereby distinguishing them from one another. One of the key advantages of CNN is its reduced reliance on extensive preprocessing compared to other classification algorithms. By utilizing relevant filters, CNN can effectively capture spatial and temporal dependencies within an image when integrated with deep learning. Consequently, it becomes adept at identifying intricate patterns in images and recognizing objects, classes, and categories, such as lung nodules in the medical field. This application enhances sensitivity in detection and reduces reading times [55], improving predictions [56], and proving invaluable in medical diagnostics [57]. Convolutional neural networks (CNN) contribute to logistics in various ways, leveraging their strengths in image recognition and classification. Some of the keyway's CNN enhances logistics operations including object detection, package sorting, etc., as we will see further on.

Computer vision stands out in the literature and employs CNN and deep learning techniques to achieve high-speed, high-volume unsupervised learning of visual information. This approach enables machine learning systems to interpret data much like the human eye does, extracting valuable insights from images and videos. Thus, computer vision aims to achieve comprehensive understanding, effectively substituting human eyes with the capabilities of computers and cameras. This technology enables the extraction of reliable measurements, precise tracking, and accurate recognition of visual elements. Given the broad scope of computer vision, it can be categorized into six well-known directions that fall under this domain: image segmentation, face recognition, object detection, image semantic segmentation, video object segmentation, and background/foreground separation [58]. The assimilation of these technical issues of computer vision within the domain of logistics enables the facilitation of tasks encompassing automated inventory management, quality control, and heightened security. Through the utilization of autonomous systems impelled by computer vision, logistical operations can attain heightened precision, augmented efficiency, and substantial cost savings over an extended duration. Each of these directions plays a significant role in extracting meaningful information from visual data, thereby contributing to the multifaceted applications of computer vision to logistics.

Image segmentation is a vital topic in the dominion of AI, which normally involves CNN and computer vision. It serves to segment or partition unidentified objects, especially those that are new or unfamiliar. Beyond this, image segmentation finds practical applications in enhancing existing methods, such as duplicate photo detection and human–computer interaction. In certain scenarios, adopting a segmentation approach brings the problem closer to semantic understanding. For instance, in content-based image retrieval, breaking down each image into smaller components when uploading them to a database allows for more refined search queries by clients. Similarly, in human–computer interaction, segmenting all elements within each video frame enables more effective interactions between the user and various persons or objects in the environment [59,60]. In logistics systems, image segmentation plays a crucial role in precisely identifying and isolating specific objects or items within an image. This capability offers benefits that enhance logistics operations. Firstly, image segmentation enables logistics companies to automate their inventory management processes. By identifying and delineating each item in stock, the system can track the quantity, location, and condition of various products. This automation streamlines inventory tracking, reduces human errors inherent in manual inventory counts, and ensures a more accurate and up-to-date view of stock levels. Additionally, image segmentation empowers logistics systems to recognize product attributes, such as shapes, colors, and sizes. This enables warehouse personnel to handle a diverse range of products. By understanding the unique characteristics of each item, the system can effectively categorize products, making it easier to organize and retrieve them. This optimization of storage and retrieval processes leads to improved warehouse efficiency and quicker order fulfillment. Overall, image segmentation is a vital component in logistics, enabling accurate identification and isolation of specific objects or items within images. Its automation capabilities contribute to more efficient inventory management, better recognition of product attributes, and optimized utilization of storage space in warehouses. By leveraging image segmentation, logistics companies can achieve higher operational efficiency, reduce costs, and deliver improved services to their customers.

Another key aspect addressed in this context is face recognition, which leverages biometric identification techniques based on human facial characteristics. In this process, video or image data containing faces are captured with a camera, and then deep learning is employed to extract facial recognition features. The model may also include display constraints or post-processing to enhance accuracy. Deep learning, particularly convolutional neural networks, is widely utilized in the field of face recognition due to its effectiveness. Additional advancements in this area include robustness modeling of deep learning for face pose, deep non-linear face recognition technology, and face recognition in various environmental contexts [61]. These sophisticated deep learning-based

face recognition technologies have significantly advanced the capabilities of this biometric identification method. The contribution to facial recognition logistics lies in its potential applications to improve security measures, which will be referred to later in the context of video object segmentation.

Another topic addressed in the literature was object detection, a fundamental task in image understanding and computer vision, which involves identifying specific item categories, such as human faces, eyes, or animals within images or videos. It serves as the cornerstone for resolving numerous complex issues, including scene understanding, picture captioning, event detection, and activity recognition, among others. Industries spanning consumer electronics, robotic vision, security, autonomous vehicles, and human–computer interfaces heavily rely on object detection [62,63]. We identified two main types of object detection challenges. The first type involves locating a particular object, like the face of a famous actor or a renowned monument. The second type aims to identify generic objects belonging to a certain category, such as birds, dogs, and vehicles, even if they have not been seen before [62,63]. Object detection plays a crucial role in revolutionizing inventory management within logistics warehouses and distribution centers. By leveraging advanced object detection algorithms, these systems can seamlessly and automatically identify and locate various items throughout the warehouse. This capability empowers logistics teams to efficiently track inventory levels, closely monitor item movements, and effectively manage their inventory in real-time. As a result, logistics operations become more streamlined, accurate, and responsive to changing demands [64].

Image semantic segmentation, on the other hand, refers to the technique of dividing an image into specific regions and extracting relevant targets. This process is vital for image processing and analysis. Instead of individually classifying each pixel, image semantic segmentation groups pixels based on their semantic relevance. For instance, a self-driving vehicle must recognize other vehicles, pedestrians, traffic signs, pavements, and various road elements for safe navigation [65]. Semantic segmentation of images finds valuable applications in quality control processes within the logistics industry. This powerful technique enables the identification and categorization of defects or anomalies in products, packaging, or shipments with remarkable precision. By employing semantic segmentation, logistics companies can ensure that only high-quality products are dispatched to customers, significantly reducing the occurrence of returns and customer complaints. This not only enhances customer satisfaction but also streamlines operations and minimizes associated costs.

In video object segmentation, the foreground target and background region are segmented into two groups of pixels, creating an object segmentation mask. This is a crucial step in behavior identification and video retrieval. Additionally, object tracking is employed to determine the location of an object within a video, proving highly valuable in intelligent surveillance. Video object segmentation is a powerful tool that enhances surveillance and security within logistics facilities. Accurately isolating moving objects or people in video streams, significantly improves the ability to detect potential unauthorized access or security breaches. This heightened vigilance ensures a safer work environment and effectively safeguards valuable assets. By leveraging video object segmentation, logistics companies can proactively respond to security threats and maintain a high level of protection throughout their operations. On the contrary, the discourse concerning security in the context of logistics also encompasses the contemplation of potential cybersecurity threats that accompany the heightened integration of automation. These concerns primarily revolve around the susceptibility of data and operational systems to malicious cyber intrusions. Therefore, it is imperative to underscore the criticality of instituting resilient security protocols to safeguard sensitive information and mitigate the likelihood of security breaches.

Object tracking and segmentation techniques complement each other, as accurate object segmentation enhances object tracking, and vice versa. Instance-level object segmentation is popular in video processing for tasks like object recognition, video editing, and video compression. Video segmentation further categorizes video content into multiple subcategories based on specific elements, such as object demarcation, movement, color,

texture, or other visual attributes [66,67]. These advancements significantly contribute to the sophisticated handling of video content and facilitate a wide range of applications in various fields.

Background/foreground separation is also a powerful image segmentation technique performed using advanced algorithms. Its applications span various fields, including intelligent surveillance in public spaces, traffic monitoring, and industrial machine vision. In recent times, neural network-based models have been increasingly employed for background separation tasks, revolutionizing the accuracy and efficiency of this process. In logistics, this technique can be applied to various scenarios, including object recognition, tracking, and inventory management. By accurately distinguishing between foreground (objects of interest) and background, logistics systems can better understand and process the visual information captured with cameras or sensors. This can improve object identification, reduce errors, and optimize logistics operations, such as automated package sorting and warehouse management. One notable application of foreground and background separation is in enhancing the detection of non-authorized objects within facilities using X-ray images [68]. By effectively isolating the foreground (the objects of interest) from the background, this technique helps security personnel identify potential threats or unauthorized items in X-ray scans more accurately and swiftly. Consequently, it contributes significantly to bolstering security measures in sensitive environments and critical infrastructures. In logistics, this has significant implications for security measures in sensitive environments such as airports, seaports, and critical infrastructure. By precisely isolating objects in the foreground from the X-ray background, security personnel can more effectively identify potential threats or unauthorized items such as contraband, weapons, or dangerous goods hidden in cargo or luggage. This feature enhances security checks and helps prevent security breaches, smuggling, and illegal transport within logistics facilities.

All in all, deep learning plays a crucial role in revolutionizing logistics, whether in the logistics warehouse or in the outdoor context of using autonomous vehicles for making deliveries [46]. While the potential of deep learning to facilitate automated actions is evident, it is essential to acknowledge the associated risks. The level of automation achieved does not guarantee a flawless outcome, and there are inherent challenges that demand careful consideration [69]. Safety, reliability, and adaptability are paramount concerns that require addressing to ensure the successful integration of autonomous vehicles into logistics operations. It is worth noting that deep learning, despite its immense promise, remains an evolving field that requires further in-depth exploration of its applications. Continued research and development efforts are needed to enhance its capabilities, overcome limitations, and maximize its positive impact on logistics. In conclusion, the integration of deep learning into logistics, especially in the realm of warehoused and autonomous deliveries, holds enormous potential. However, a cautious approach must be taken, considering the challenges and risks associated with automation. By dedicating efforts to advancing the field through continuous investigation, the logistics industry can harness the true transformative power of deep learning.

### 3.2. Robot-Driven Logistics

The growth of autonomous vehicles, as discussed in the previous section, has been spurred by the exponential rise of electronic commerce (E-Commerce), leading to a significant increase in the daily volume of goods being transported. Consequently, consumers have developed higher expectations for deliveries, demanding speed, and convenience [70]. To meet these demands, autonomous electric distribution and delivery vehicles are revolutionizing the industry by reducing human labor and enabling quicker deliveries, resulting in a 25% to 35% increase in the number of successful deliveries. As automation becomes more prevalent in logistics, the challenge lies in enhancing decision-making capabilities for autonomous robot navigation, which requires complex self-optimization and self-configuration processes [19,71]. Autonomous robots have the potential to forecast delivery schedules for electric vehicles, leading to a significant reduction in faults and errors during

the delivery process [70]. As this technology continues to advance, it will pave the way for even more efficient and reliable delivery systems in the future.

Therefore, AI is playing a vital role in enhancing safety and building trust in robots [72]. Traditionally, robots were perceived as products of mechanical or electrical engineering, adopting a bottom-up approach that prioritizes embodiment and sensorimotor functions. However, the integration of AI with robotics has led to remarkable innovations [73]. One area in which AI and technology have improved internal logistics is reducing errors during product picks in warehouse stations. Utilizing Radio Frequency Identification (RFID) technology has proven effective in mitigating these errors [74]. RFID chips, also known as integrated circuits (ICs), store unique identifiers transmitted via radio waves to nearby readers, making it possible to locate and track autonomous robots with ease [75]. Companies like Confidex have specialized in developing RFID tags designed to withstand chemicals, washing, and high temperatures during the manufacturing process, further automating logistics [76]. Despite the advantages of RFID, challenges remain, including malfunctioning tags and the time-consuming creation and maintenance of look-up tables with geo-positions [77]. Here, computer vision offers a valuable solution as it serves as the primary sensory system for both navigation and load detection [78]. Companies like DHL leverage computer vision by employing cameras to capture images or videos, and AI algorithms to analyze the data from the digital imagery. These AI systems can distinguish between items and even learn to follow objects across various views autonomously [79]. By combining the strengths of AI and robotics, we can continue to enhance the efficiency, reliability, and safety of robotic systems, ensuring a bright future for autonomous technologies.

However, not everything is perfect in relation to humans and technologies such as AI and robots, as risks exist. For instance, in this regard, Oliff et al. [80] emphasize that reinforcement learning (RL) plays a crucial role in enhancing human–robot interaction, especially in autonomous robotics operations within internal logistics, but security risks remain a significant concern. Kiangala et al. [81] support this perspective by highlighting RL as a valuable tool that enables robots to learn new paths and navigate complex and unpredictable environments without prior knowledge, thus reducing potential dangers. A collaborative approach in human–robot integration (HRI) opens possibilities for developing social service robots that coexist harmoniously with humans in their social environments, effectively meeting the needs and expectations of lay experts. However, designing service robots that are relevant to human social contexts while aligning with popular media portrayals and preparing for the future of technology poses more challenges [82].

As the number of humans in smart factories remains high, unpredictability and disturbances, such as data misplacement, miscommunication, and process skipping, persist. To address this, robotic operations must adapt their behavior to accommodate variations in human task performance [80]. Currently, robots primarily react to pure sensor information regarding safety requirements, resulting in inadequate responses when encountering unexpected obstacles [83]. A significant drawback is that robots fail to halt their activities in the presence of unforeseen events or when humans approach the safety perimeter, as laser scanners lack the necessary information, such as 3D size, distance, or direction of objects. Implementing sensor elements like time-of-flight or radar sensors provides an effective solution by enabling the detection of 3D environmental information [83]. Knight [84] argues that human–robot interaction entails a balanced combination of profound comprehension of information, mechanical capabilities, sophisticated system thinking, and the ability to handle novel or unexpected phenomena while considering human interests. He also notes that current robotic companies tend to prioritize the perception of perfection over promoting open communication and collaboration within their ecosystems. Additionally, the lack of information regarding types of robot failures and potential solutions poses a challenge. Existing robotic fail-handling paradigms struggle to address foreseeable failures, let alone unforeseen ones that can disrupt the human–robot ecosystem (HRE) [85]. The graceful extensibility theory offers a framework to model, assess, and enhance the capacity of HRE to respond to unforeseen failures. By extending the human–robotic interaction

(HRI) to the HRE, adaptive fail-handling strategies and the social and technical enablers to support these strategies can be identified [85]. For a summary of this section, refer to Figure 2 below.

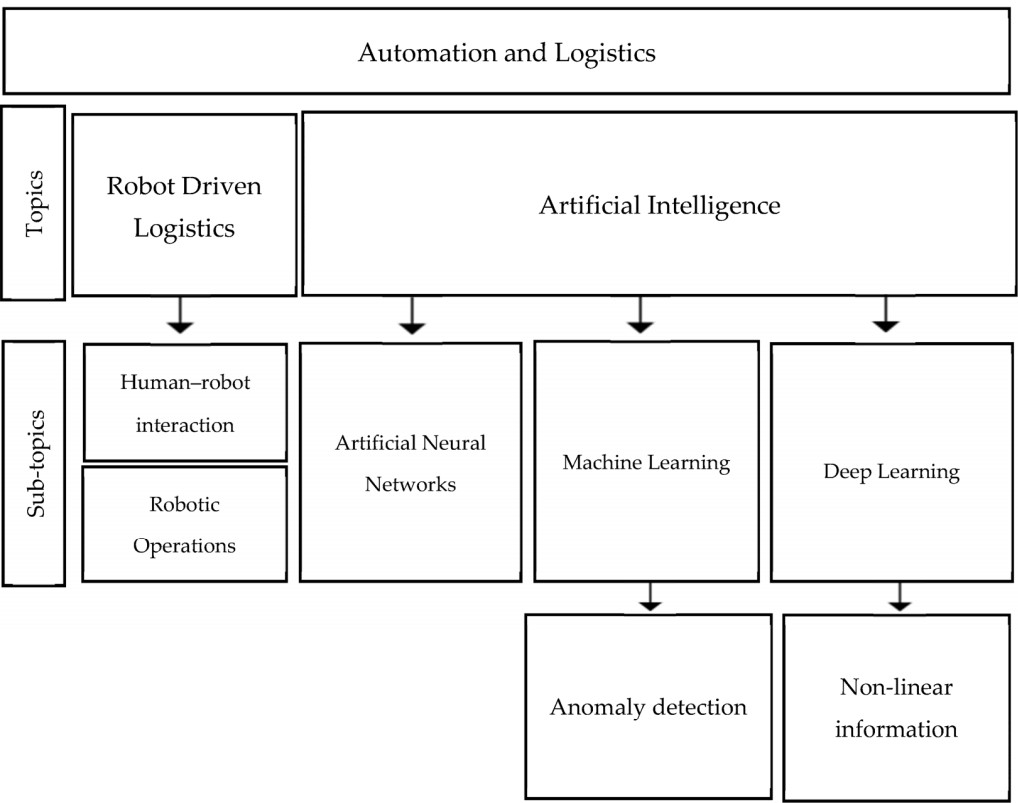

**Figure 2.** Emerging topics of automation in logistics.

## 4. Conclusions

### 4.1. Theoretical and Managerial Contributions

This research makes significant theoretical contributions by providing novel perspectives on the application of automation in logistics. While there are existing studies on the topic, this research stands out for its unique analysis of how automation is currently enhancing the logistics industry, offering a more comprehensive understanding compared to the existing literature. Employing the PRISMA statement allowed us to focus our search and highlight the most crucial themes.

Our research revealed that artificial intelligence and robot-driven logistics have a profound impact on how automation supports logistics operations, resulting in improved competitiveness and a reduced margin of error. The potential of automation-based solutions in logistics has captured the attention of managers and practitioners across various logistics businesses and sectors. However, due to their complexity, practitioners often struggle to assess their practical use or gain a holistic understanding of the current landscape. Therefore, our aim was to bridge this gap by presenting real-world examples from the management literature, which can serve as instructive references for both academics and practitioners. By offering a summary of available approaches and fresh ideas for planning and implementing automation initiatives in logistical processes, this research equips practitioners with valuable insights to navigate the challenges and leverage the benefits of automation effectively.

Hence, it is the contention of this research that the implications of the findings hold wide-ranging significance and bear substantial potential for various stakeholders. Through an extensive exposition of the underlying phenomenon, this study delineates its primary implications and prospective benefits. In essence, our aim is to target four key cohorts.

Firstly, among those mentioned in the previous paragraph, professionals and businesses can gain from the acquisition of valuable insights concerning the diverse applications of automation technologies within their operations. By leveraging these insights, enterprises can make well-informed decisions about the assimilation of automation into their logistics frameworks, thereby fostering enhanced performance and adaptability within the swiftly evolving logistics landscape.

Simultaneously, scholars and researchers can embark on a trajectory akin to our own, exploring the convergence of automation and logistics. By employing the findings, researchers can delve deeper into the dynamics of human–robot interaction and explore the scope for the fusion of artificial intelligence with robotic systems in the purview of logistics processes. Alternatively, they may pursue our recommendations for prospective research, such as the empirical validation of the conceptual framework or the adoption of quantitative methodologies. Thus, this study lays the groundwork for more profound analyses, whether quantitative or empirical, and sets the stage for pioneering research endeavors that can catalyze transformative changes within the field.

Thirdly, policymakers can glean meaningful insights from this research, which can serve as a compass for the formulation of guidelines governing the integration of automation in the logistics sector. By comprehending the potential benefits, challenges, and best practices associated with the adoption of automation, policymakers can devise strategies that advocate for the responsible and ethical implementation of automation technologies in logistics operations. Addressing the ethical ramifications of automation within the logistics sector, particularly concerning job displacement and the ethical application of AI in decision-making procedures, holds significant importance. Consequently, advocating for the conscientious deployment of AI and acknowledging the social repercussions of automation on the labor force and the broader society emerges as a pivotal consideration for policymakers.

Lastly, the general populace and consumers stand to benefit from the integration of automation in logistics, which can engender improved service provision, expedited deliveries, and enhanced consumer experiences. By enabling more efficient and reliable logistics operations, automation can contribute to the punctual and accurate conveyance of goods and services to consumers, thereby fostering greater satisfaction and cultivating stronger customer relations. Furthermore, the study underscores the criticality of upholding a balance between automation and human engagement, accentuating the potential for automation to augment, rather than supplant, human labor within the logistics industry. By underscoring the benefits, challenges, and best practices associated with the integration of automation into logistics, this study sets the stage for future advancements and innovations within the domain.

### 4.2. Research Limitations and Suggestions for Future Research

The methodologies employed in this study bear responsibility for their limitations. Systematic reviews, while valuable, offer a moment-in-time snapshot of a specific reality [86]. Consequently, despite updating scientific databases with fresh, peer-reviewed research, the information eventually becomes outdated [87]. Our approach of limiting the search to specific terms and using filters to select significant articles carried the risk of omitting relevant manuscripts. Despite these acknowledged limitations, we believe that this systematic review holds value as it provides a concise overview of the current state of automation in logistics. It is important to acknowledge that the rapidly evolving nature of technology and logistics means that new advancements and innovations are continually emerging. Therefore, our study, although comprehensive within its scope, may not capture the very latest developments in the field. To address this limitation, we encourage researchers and practitioners to continually update and expand on our findings, incorporating new data and insights as they emerge. The results of this study should serve as a catalyst for further research on the application of automation in the field of logistics within industrial enterprises. We believe that this research can lay the groundwork for

more in-depth investigations into specific aspects of automation, such as its impact on supply chain efficiency, cost-effectiveness, and environmental sustainability. Furthermore, exploring human–robot interaction dynamics and the potential for integrating artificial intelligence with robotic systems can open new avenues for optimization and innovation in logistics processes. Moreover, the results offer industrial company practitioners a valuable framework for effectively implementing cutting-edge technologies. As automation continues to revolutionize the logistics industry, organizations can utilize our findings to be more informed and shape their automation strategies. By understanding the potential benefits, challenges, and best practices associated with automation adoption, businesses can proactively adapt to the changing landscape and gain a competitive advantage. In conclusion, while this study has its limitations, we believe it helps to provide a deeper understanding of automation in logistics. As the field continues to evolve, we encourage fellow researchers and industry practitioners to build upon this work, pushing the boundaries of knowledge and innovation to unlock the full potential of automation in revolutionizing industrial logistics. To illustrate, scholars can fortify these findings through a quantitative/bibliometric analysis of the literature, elucidating the essential components of logistics automation, such as the significance of each technology, frequency of citation for individual techniques, keyword tracking, and so on. On the other hand, practitioners can contribute by identifying recent practical cases that facilitate the empirical validation of the conceptual framework proposed in this article. By collaboratively advancing research in this domain, we can drive transformative changes that lead to more efficient, sustainable, and resilient logistics operations in the modern era.

**Author Contributions:** Conceptualization, B.F.; methodology, B.F. and J.R.; software, J.R.; validation, J.R.; formal analysis, B.F.; investigation, B.F. and J.R.; resources, B.F. and J.R.; data curation, B.F. and J.R.; writing—original draft preparation, B.F.; writing—review and editing, J.R.; visualization, B.F. and J.R.; supervision, J.R.; project administration, B.F. and J.R.; funding acquisition, J.R. All authors have read and agreed to the published version of the manuscript.

**Funding:** This research received no external funding.

**Institutional Review Board Statement:** Not applicable.

**Informed Consent Statement:** Not applicable.

**Data Availability Statement:** To access the data referenced in this article, please reach out to the corresponding author for further information.

**Acknowledgments:** We express our sincere gratitude for all the support extended to researchers by ISLA-Santarém.

**Conflicts of Interest:** The authors declare no conflict of interest.

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
