# Peer review of "A Systematic Literature Review on the Application of Automation in Logistics"

_logistics_

Round 1
Reviewer 1 Report
Comments and Suggestions for Authors
In the abstract, please inform what are the results and the main implications of the study;
Please strictly avoid lumped references [3-6], as for other researchers, it would be useful to individuate what study offers each contribution;
You have a strong flaw in your study. You state that … While automation has been extensively studied in engineering, marketing, and medicine [3-6], logistics in scientific research still falls short when compared with these domains. … Why should I believe in it? You must provide evidence to support such a statement. The way you should take is to lead a search on databases (Scopus is OK) and demonstrate that [logistics + AI] is less cited than [engineering + AI], [marketing + AI], and [medicine + AI] in the last five years;
The application of the PRISMA protocol is not clear in the last step. Please clarify a little bit more why the samples passed from n = 17 to n = 29;
There is a gap between your sections 2 and 3. In section 2, you applied the PRISMA protocol to identify 29 (I believe there are 19) articles. In section 3, you describe the main domains in automation. It lacks a bridge among them. How can I be sure that you identified the right domains? A bibliometric analysis of the articles' content should help identify what is and what is not significant in logistics automation.
The contribution of section 3 is limited. It is only a review of topics related to automation in logistics. It does not bring a perspective view of the importance of each technology, how many times each technique was cited, keywords tracking, etc. The same is true regarding Figure 2. If you replace the expression “Automation and Logistics” with, for instance, “Automation and Schedule”, or “Automation and Forecasting”, the figure also applies. You should state the relative importance or percentage of keywords cited in each technique and identify clues for further research. As is, the figure has little utility.
In the last section, please introduce a discussion on the implications and further steps of your research. Who may gain what and why upon your study?
Best regards
Reviewer 2 Report
Comments and Suggestions for Authors
Define Scope Clearly: Specify the exact scope of the paper in terms of automation in logistics. What aspects of logistics are covered, and what are excluded? Providing a clear scope will help readers understand the paper's focus.
Quantify Growth: When discussing the growth of automation in various sectors, consider including statistics or growth percentages to support your claims. This adds quantitative evidence to your argument.
Provide Concrete Examples: While discussing automation's potential, offer specific examples of how AI, machine learning, and deep learning have been applied in logistics to enhance efficiency. Real-world case studies can make your points more tangible.
Technical Details: Elaborate on the technical aspects of automation. For instance, explain how AI algorithms are used for autonomous task execution in logistics. Provide technical insights into how self-problem resolution works.
Discuss Challenges and Risks: Address potential challenges and risks associated with automation in logistics. This could include cybersecurity concerns, ethical considerations, or the need for human oversight. Acknowledging challenges demonstrates a balanced perspective.
Clarify "Smart Logistics": Define what you mean by "smart logistics" and how automation contributes to making logistics "smart." This term may have different interpretations, so providing a precise definition is essential.
Please avoid citing sources that were published before to 2019. Cite current research that are really pertinent to your topic. The study also lacks sufficient citations. Another critical step is to compare the topic of the article to other relevant recent publications or works in order to widen the research's repercussions beyond the issue. Authors can use and depend on these essential works while addressing the topic of their paper and current issues.
Amiri, Z., Heidari, A., Navimipour, N.J. et al. Adventures in data analysis: a systematic review of Deep Learning techniques for pattern recognition in cyber-physical-social systems. Multimed Tools Appl (2023). https://doi.org/10.1007/s11042-023-16382-x
Heidari, A., Navimipour, N. J., & Unal, M. (2022). Applications of ML/DL in the management of smart cities and societies based on new trends in information technologies: A systematic literature review. Sustainable Cities and Society, 104089.
Nitsche, Benjamin. "Exploring the Potentials of automation in logistics and supply chain management: Paving the way for autonomous supply chains." Logistics 5.3 (2021): 51.
Alnahas, Jasim. "Application of Process Mining in Logistic Processes of Manufacturing Organizations: A Systematic Review." Sustainability 15.15 (2023): 11783.
Comments on the Quality of English Language
Moderate editing of English language required
Reviewer 3 Report
Comments and Suggestions for Authors
The manuscript is basically a review article. Need to produce study based on lots of recent relevant literature. More precisely the recent challenges in the thrust area and opportunities for future research.
With reference to the automation logistics, in recent years lots of Industry 4.0 technologies have significantly contributed . Please improve the literature and expand the future scope. The following are certain keywords will be helpful in identifying recent literate.
“AR/VR technologies for logistics” “Challenges and opportunities in human robot collaboration”; “Challenges and opportunities on AR/VR technologies for manufacturing systems” “Managing uncertainty in logistics”
Also please address, how automation can address the uncertainties in the logistics & support.
Give an illustration (graphical) or a table to concise the information given in section 3.
Besides the technical detail, please check the formatting errors, please refer the images (Figure 2)
Reviewer 4 Report
Comments and Suggestions for Authors
The paper presents a systematic review on automation in logistics. The topic is interesting and beneficial for the completion of the database of scientific knowledge in the field of technologies used in logistics. The theme is also important in terms of predicting the further development of technologies used in logistics, as the authors have indicated in the paper.
I consider the processing and documentation of the survey to be problematic. The authors refer to PRISMA Statement, which they used for the systematic review. However, there are many inconsistencies in the paper. For example:
- The PRISMA flowchart in Figure 1 and the accompanying text in the section 2.2 do not correspond. The diagram shows 17 sources assessed for eligibility. It means positively assessed (according to PRISMA). However, the authors stated 17 excluded articles in the text.
- The PRISMA Statement refers to source number 13 (p.2, line 77), which dates from 2003. However, PRISMA was introduced in 2009.
- The authors state that the review only considered articles published between 2019 and 2023. The Results also include information and cite sources much older than 2019. Additionally, nowhere is what sources have been considered for the review, as a lot more than 26 are cited in Results. The articles used for the systematic review should be listed in some way.
- It is claimed that articles other than journal papers were excluded. But there also were many conference papers used (e.g., sources 53, 54.).
- In Fig.2, the authors list Artificial Neural Networks as a separate topic, which is not treated as a separate subsection in the review as other topics. It is a bit confusing.
- In the last step of the research, other articles were included in the study, offering relevant contributions. Nowhere has been stated their number and what was the criterion for their selection (Other database, type of article, period?).
I recommend describing the research according to the PRISMA Statement to be clear and comprehensive.
Round 2
Reviewer 1 Report
Comments and Suggestions for Authors
Ok
Reviewer 2 Report
Comments and Suggestions for Authors
No comment
Reviewer 3 Report
Comments and Suggestions for Authors
The manuscript is considerably revised and the responses of the reviewers are satisfactory.
No further comments to the authors.